# Contrast-Enhanced Black Blood MRI Sequence Is Superior to Conventional T1 Sequence in Automated Detection of Brain Metastases by Convolutional Neural Networks

**DOI:** 10.3390/diagnostics11061016

**Published:** 2021-06-01

**Authors:** Jonathan Kottlors, Simon Geissen, Hannah Jendreizik, Nils Große Hokamp, Philipp Fervers, Lenhard Pennig, Kai Laukamp, Christoph Kabbasch, David Maintz, Marc Schlamann, Jan Borggrefe

**Affiliations:** 1Institute for Diagnostic and Interventional Radiology, University Hospital Cologne, Kerpener Strasse 62, 50935 Cologne, Germany; h.jendreizik@gmail.com (H.J.); nils.grosse-hokamp@uk-koeln.de (N.G.H.); philipp.fervers@uk-koeln.de (P.F.); lenhard.pennig@uk-koeln.de (L.P.); kai.laukamp@uk-koeln.de (K.L.); christoph.kabbasch@uk-koeln.de (C.K.); david.maintz@uk-koeln.de (D.M.); marc.schlamann@uk-koeln.de (M.S.); 2Division of Cardiology, Pneumology, Angiology and Intensive Care, University of Cologne, Kerpener Strasse 62, 50935 Cologne, Germany; simon.geissen@uk-koeln.de; 3Institut für Radiologie Neuroradiologie und Nuklearmedizin, Ruhr-Universität Bochum, Minden University Hospital, Hans-Nolte-Strasse 1, 32429 Minden, Germany; Jan.Borggrefe@muehlenkreiskliniken.de

**Keywords:** magnetic resonance imaging, convolutional neural networks, automated detection of brain metastases

## Abstract

Background: in magnetic resonance imaging (MRI), automated detection of brain metastases with convolutional neural networks (CNN) represents an extraordinary challenge due to small lesions sometimes posing as brain vessels as well as other confounders. Literature reporting high false positive rates when using conventional contrast enhanced (CE) T1 sequences questions their usefulness in clinical routine. CE black blood (BB) sequences may overcome these limitations by suppressing contrast-enhanced structures, thus facilitating lesion detection. This study compared CNN performance in conventional CE T1 and BB sequences and tested for objective improvement of brain lesion detection. Methods: we included a subgroup of 127 consecutive patients, receiving both CE T1 and BB sequences, referred for MRI concerning metastatic spread to the brain. A pretrained CNN was retrained with a customized monolayer classifier using either T1 or BB scans of brain lesions. Results: CE T1 imaging-based training resulted in an internal validation accuracy of 85.5% vs. 92.3% in BB imaging (*p* < 0.01). In holdout validation analysis, T1 image-based prediction presented poor specificity and sensitivity with an AUC of 0.53 compared to 0.87 in BB-imaging-based prediction. Conclusions: detection of brain lesions with CNN, BB-MRI imaging represents a highly effective input type when compared to conventional CE T1-MRI imaging. Use of BB-MRI can overcome the current limitations for automated brain lesion detection and the objectively excellent performance of our CNN suggests routine usage of BB sequences for radiological analysis.

## 1. Introduction

Representing the largest proportion of brain neoplasms, brain metastases pose a key issue in cancer follow-up or tumor staging [1,2]. Early detection and treatment of small brain metastases (<15 sq mm; <5 mm short axis diameter (SAD)) has severe implications for therapy and may significantly prolong survival of cancer patients [3,4]. Accurate detection of small metastases may be a difficult and tiresome task, even for experienced radiologists, especially given the rising number of screening examinations for metastases and the introduction of 3D scans [5,6].

Convolutional neural network (CNN) based deep learning bears the potential to detect pathological changes even more sensitively than trained human experts [7,8,9]. In clinical routine, CNN are already routinely used in CT examinations showing an increase of the detection rates for lung nodules [10]. However, CNN are not established for the detection of brain metastases, even though there are several CE certified start-ups and a number of studies in literature reporting CNN performance for this task on CE T1 sequences [11,12]. In contrast to conventional CE T1 imaging, MRI black blood imaging (BB) reduces intraluminal blood signal over the field of view without decreasing the signal intensity of small metastases, thus facilitating detection of brain lesions for human readers (see Figure 1) [5,13,14,15,16].

Several other studies approached the computer aided detection of brain neoplasms in recent years. A 3D selective enhancement filter for the detection of brain metastases and different template matching-based algorithms have been proposed [6,17,18,19]. CNN were successfully used for automatic detection and segmentation of meningiomas and gliomas and T1 3D based recognition of metastases [7,8,20,21,22]. Yet, automated detection of brain metastases poses an extraordinary challenge, especially due to small lesions the size of brain vessel diameters, but also due to other contrast enhancing structures, such as the tentorium, plexus or vessel anomalies such as hypertensive vessel changes or aneurysms, resulting in high false positive rates [5,9,16].

However, current computer aided recognition systems are not sufficiently detailed on T1 gradient-echo or CE T1 sequences, resulting in high rates of false positive results and, thus, the questioning of their usefulness in clinical routine [7,9,17,18,19,23]. In contrast, Yang et al. proved BB-based automatic detection of brain metastases using computer aided recognition systems to be of superior sensitivity compared to MP-RAGE T1 CE [24]. However, this study was limited to 29 patients and facilitated an artificial neural network after the application of feature extraction. To the best of our knowledge, our study is the first to evaluate the performance difference of CNNs trained on CE T1 and BB data that does not facilitate image segmentation or feature extraction. Whereas application of these methods may result in an improved performance, specifically on limited datasets, the usage of intense preprocessing and specific annotation strategies may limit the reproducibility of the study outcome.

Given the aforementioned limitations of conventional CE T1 sequences, we postulate differences in the efficiency of conventional CE T1 and BB sequences as input data for CNN training with potential advantages for BB sequences in terms of automated CNN-based detection. For this reason, we compared the performance of a CNN for the detection of brain lesions trained on either BB or CE T1 spin echo (CE T1) MRI images.

## 2. Materials and Methods

In accordance with IRB guidelines, the preparation of this retrospective, single center study did not require an ethics vote.

### 2.1. Patient Population

Using our institutional image archiving system, MRI scans of patients receiving screening MRI for brain metastases at our tertiary-care university hospital between June and October 2017 were reviewed. Scans included, when patients received both, BB and conventional CE T1. Accordingly, 127 patients were identified. Scans were excluded given following reasons: (I) prior neurosurgical procedures (*n* = 32), (II) number of metastases (>8; *n* = 4), and (III) meningeal carcinomatosis (*n* = 6). This resulted in a total number of 85 patients, of which 26 presented a total number of 47 brain metastases. The remaining healthy 59 patients served as a control group. Labeling of the image dataset was based on the original radiological report and on additional manual sighting of all included slides. The sizes of metastatic lesions included in our study are documented in Figure 2.

### 2.2. Imaging

Image acquisition was performed using two 3 Tesla Ingenia Scanners (Philips Healthcare, Best, The Netherlands). The standardized imaging protocol at the university hospital of Cologne included intravenous administration of gadolinium (Dotarem, Guerbet GmbH, Roissy, France; 0.5 mmoL/mL, 1 mL = 279.3 mg gadoteric acid = 78.6 mg gadolinium) with a concentration of 0.1 mmol/kg body weight into an antecubital vein. 

The acquisition matrix was 404 × 254 × 30 for conventional CE T1 and 404 × 254 × 410 for BB. Voxel size was 0.62 × 0.75 × 5 mm for conventional CE T1 (T1-FFE) and 0.8 × 0.8 × 0.8 mm for BB. Repetition time was 320 ms for conventional CE T1 and 700 ms for BB. Echo time was 4.61 ms for conventional CE T1 and 35 for BB, using a flip angle of 80° in both sequences. BB acquisition was TSE-based and the echo train length was 1.0.

### 2.3. CNN

A CNN with weights pretrained on the ImageNet database (Inception V3 with ImageNet weights, Google, Mountain View, CA, USA) was retrained using the TensorFlow framework (Google, Mountain View, CA, USA) with a customized monolayer classifier, using either conventional CE T1 or BB scans of brain lesions and physiological brain scans without additional annotation (e.g., bounding boxes). For every patient presenting brain lesions, we exported every 2D MRI axial slide depicting the lesions to the input and discarded the remaining slices. Each lesion included in the learning process had both an identical BB and a CE T1 image available.

For the control group, complete slide sets of 59 healthy patients were included (total number of included slide images: *n* = 43 lesion slides, 545 control slides for CE T1 and *n* = 88 lesion slides, 1306 control slides for BB). This ensured that every potential layer with a lesion had a corresponding one without lesion available for the learning process (see Figure 3). To assimilate the size of training sets in both experimental groups, after formation of a holdout validation set containing 10% of the original images, we facilitated image augmentation by randomly flipping and rotating up to 20°, thereby creating a total of *n* = 274 lesion slides, 981 control slides (CE T1) and *n* = 316 lesion slides, 1175 control slides (BB) images. For augmentation, the Python package imgaug was used (Python 3.7; source code and documentation accessible at https://github.com/aleju/imgaug, accessed on 30 May 2021).

During training, the datasets were randomly separated into training and validation sets, with a validation rate of 0.2. After creating bottlenecks for all images using the pretrained Inception v3 network, an additional classifier was trained on top of the network using a total of 1k training steps. Every ten steps, validation accuracy was determined to estimate learning progress and check for overfitting.

After training, the resulting graphs were applied to the holdout validation set using label images with a predictive value representing the certainty of class association. 

### 2.4. Statistical Analysis

Statistical analysis and preparation of the graphs were performed using R version 3.6.2 (https://cran.r-project.org/, accessed on 30 May 2021). By variation of classification threshold, sensitivity and specificity curves, as well as AUC, were calculated and visualized using the pROC package (documentation available at https://cran.r-project.org/web/packages/pROC/pROC.pdf, accessed on 30 May 2021). Graphics and figures were created using the ggplot package. The P threshold was set in two-sided t-tests to 0.05.

## 3. Results

The total number of patients was 85, of which 26 presented a total number of 47 brain metastases. Of these, 12 were related to non-small cell lung carcinoma (NSCLC; 25.5%), 24 to malignant melanoma (51%), four to hematological neoplasia (8.5%), one to renal cell carcinoma (2.1%), three to prostate carcinoma (6.4%), and three to breast carcinomas (6.4%). The average patient age was 59.9 years (SD 17.2), with 52.3 % of the patients male and 47.7 female.

CE T1 imaging-based training resulted in a final internal validation accuracy of 85.5% after 1 k training steps (see Figure 4a,b,c), whereas BB image retraining showed a final internal validation accuracy of 92.3% (*p* < 0.01).

In holdout validation analysis, CE T1 image-based prediction presented poor specificity and sensitivity with an AUC of 0.534, whereas the BB image-based prediction model performed with an AUC of 0.869 (see Figure 5). Furthermore, it was revealed that, in particular, smaller metastases (<15 sq mm; <5 mm SAD) can be detected better using BB scans in the internal training data set (*p* < 0.05; see Figure 4d) as well as in the holdout validation data set.

## 4. Discussion

In this study, we investigated whether conventional CE T1 or CE BB sequences provide the input data of choice for computer-aided detection of brain metastases.

As major finding of this study, CE BB MRI imaging proved to be a significantly more effective input type for CNNs compared to traditional CE T1 MRI imaging, resulting in higher specificity and sensitivity performance.

Comparing the present, methodologically different approach with pre-processed data from Yang et al., both show relatively similar values of an AUC of approximately 0.9 with respect to the BB data set and a slightly smaller AUC with respect to the conventional T1 CE approach in this study. This is an effect that may be explained by the different number of patients. In summary, we agree with the results of Yang et al. and show that even with a new concept of using non-preprocessed data, the BB-based approach of automated metastasis detection is clearly superior to the approach based on conventional T1 CE data.

For smaller lesions under 5 mm SAD, BB data proved to be superior to conventional CE T1 and should primarily be useful for automated detection of brain metastases. BB data thereby emerges as an alternative modality for training algorithms specialized in high-sensitivity detection of very small brain lesions, a repeating issue even with otherwise excellently performing machine learning algorithms [5,6]. Beyond this, the objectively quantified performance of our CNN on BB sequences also suggests the routine usage of BB for radiological analysis.

This study is limited because it was conducted in a single-center using a retrospective approach and should be considered as a proof-of-concept for both the superiority of BB imaging data in training radiological classifiers and the application of pre-trained CNNs for radiological image data.

Since our study focused largely on a potential difference between T1 CE and BB imaging data and their potential as input for neural network training, we decided to facilitate a CNN pretrained on a very large dataset and commonly used for transfer learning applications. Although a CNN pretrained on radiological data might provide better absolute performance and should, in future, be explored for clinical application, the obvious differences between both acquisition types in a CNN pretrained on non-radiological data clearly underline the performance gain from using BB data over traditional T1 CE for metastasis detection.

Of importance, other than a previous study that used BB imaging for automatic brain lesion detection, no segmentation of images or feature extraction was performed; instead, the CNN was trained in an unsupervised manner and for the most part used weights obtained from training on non-radiological image data. The excellent performance of the CNN encourages future research to use such pretrained networks that profit from unsupervised learning on huge datasets.

There are important limitations to our study. Since the encoding of T1 CE and BB CE datasets is different, the analyzed 2D slices may differ in artifact frequency and specifically, and BB images may contain information that is lost in the inter-slice gaps of the T1 CE datasets. Furthermore, the voxel size between T1-CE and BB-CE dataset was different, which could potentially impact detectability of the lesions, although only images where the lesion was clearly visible for human experts were included in our study. Additionally, the feasibility of performing BB CE imaging during every MRI brain scan for metastasis detection is questionable as this technique prolongs image acquisition significantly. Of importance, computational solutions for artificially creating BB-like images from traditional T1 and T2 sequences are emerging, proving that it may not be necessary to natively acquire BB imaging to benefit from its improved distinctiveness of vessels and metastatic lesions [25]. Further, CNNs may integrate information from multiple acquisition methods to further increase classification performance in the future.

Although this study represents the largest study known to us in the field of automated brain metastasis detection using black blood sequence, automated detection using neuronal networks should always aim for the largest possible data sets. Larger data sets with both sequences could further increase the detection rate and increase specificity and sensitivity and would allow implementation of a more customized CNN architecture. Further studies will provide knowledge about sequence-typical errors of omissions and commission by means of a superordinate decision network, and, in this way, optimize clinical routine and radiological reports. 

## 5. Conclusions

In conclusion, this work showed the superiority of CE BB over conventional T1 sequences for the automated detection of brain metastases using CNN. As CNN-based deep learning bears the potential to detect pathological changes with improved sensitivity in comparison to trained human experts, it will soon be implemented in clinical routine [5,16]. These findings prove to be of high relevance, as the use of AI for brain lesion detection is to be an expected standard application in clinical routine.

## Figures and Tables

**Figure 1 diagnostics-11-01016-f001:**
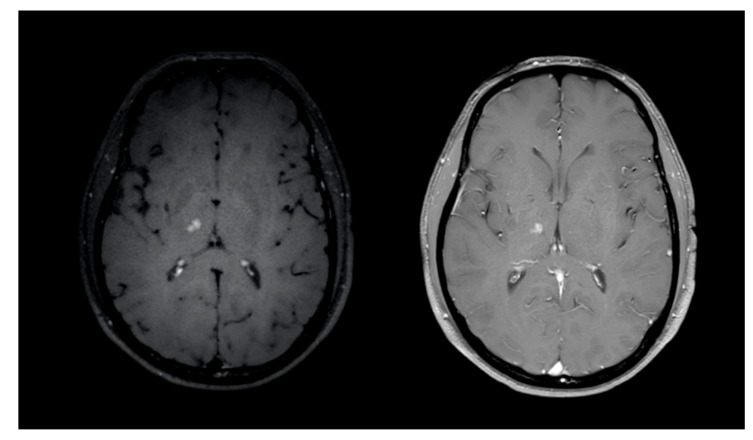
**Left**: lesion in the black blood sequence. **Right**: corresponding axial conventional CE T1 slice with identical lesion.

**Figure 2 diagnostics-11-01016-f002:**
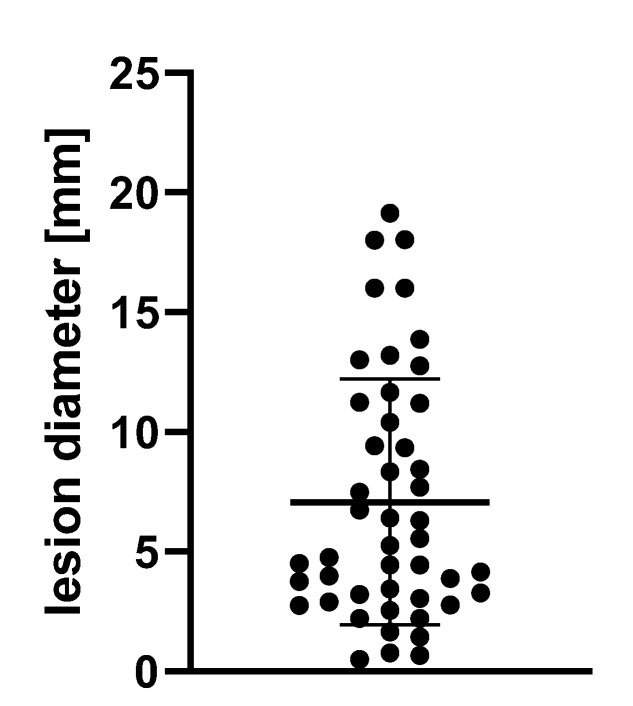
Diameter of analyzed lesions.

**Figure 3 diagnostics-11-01016-f003:**
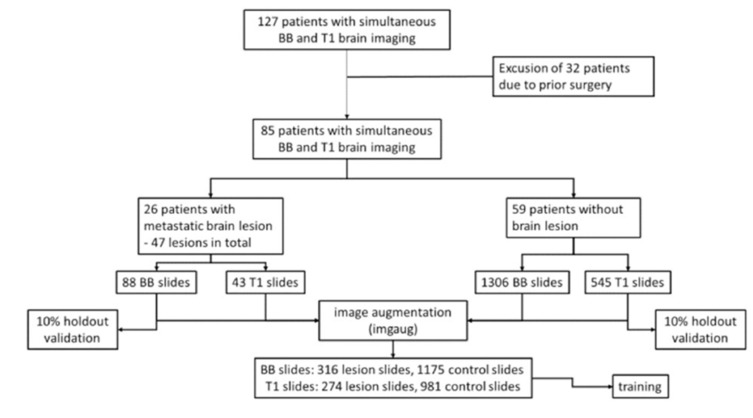
Flow chart for patient selection and split of the data set.

**Figure 4 diagnostics-11-01016-f004:**
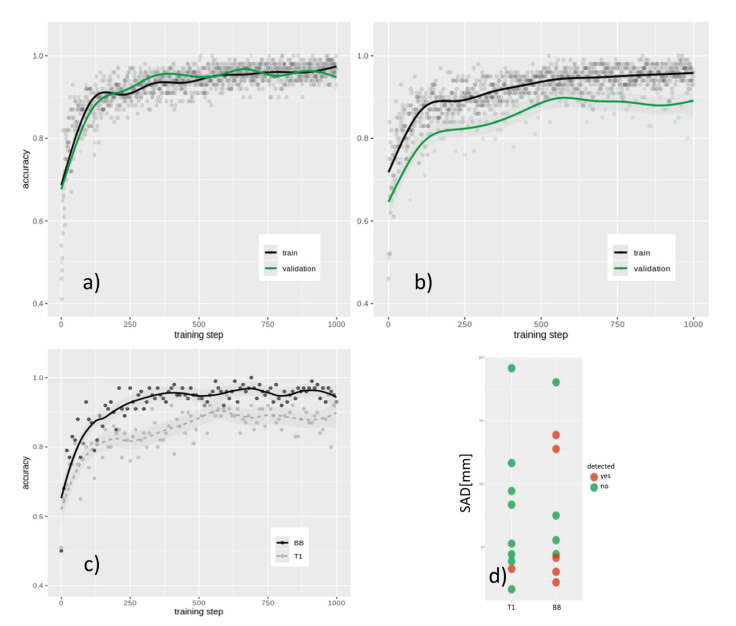
Training and validation accuracy of the (**a**) BB and (**b**) conventional T1 data sets; (**c**) validation accuracy of both data sets in comparison; (**d**) metastasis size of the detected lesion in the holdout validation analysis by corresponding set.

**Figure 5 diagnostics-11-01016-f005:**
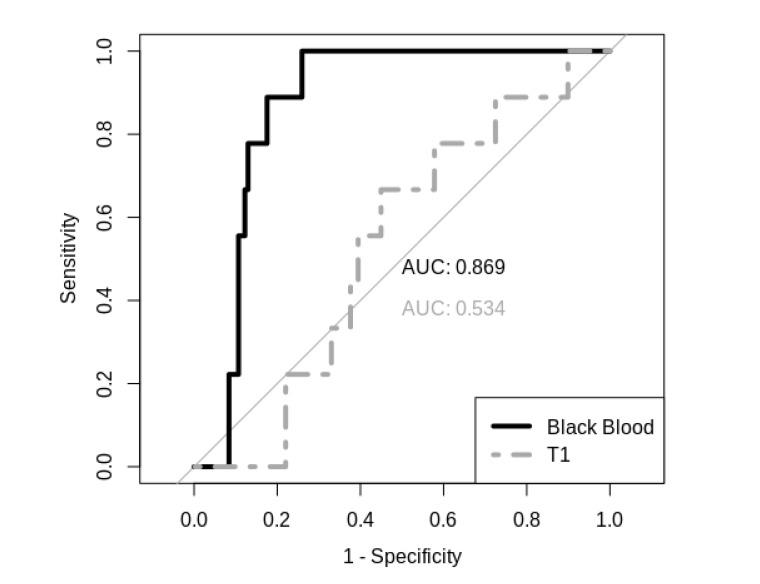
Sensitivity and specificity curves (BB in black, conventional T1 in grey).

## Data Availability

The data presented in this study are available on request from the corresponding author. The data are not publicly available since these are admittedly completely anonymized patient data in clinical context.

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
