# Peer review of "Contrast-Enhanced Black Blood MRI Sequence Is Superior to Conventional T1 Sequence in Automated Detection of Brain Metastases by Convolutional Neural Networks"

_diagnostics, 2021, doi:10.3390/diagnostics11061016_

Round 1

Reviewer 1 Report

The work “Contrast-Enhanced Black Blood MRI Sequence is Superior to Conventional T1 Sequence in Automated Detection of Brain Metastases by Convolutional Neural Networks” explores the possibility of enhancing the convolutional neural network performance in metastases detection via modifying the data acquisition protocol. The study shows that a blood-suppressed imaging sequence provides better performance metrics over conventional 3D T1-weighted image. This, in a way, serves as quantitative evidence of previously detected effect (https://pubmed.ncbi.nlm.nih.gov/22104961/). As a principal difference to previous works lies in the use of CNNs, a number of methodological choices need to be addressed, primarily concerning the way CNNs are used.

  • Please provide the reasoning (particularly, benefits and possible downsides) of using a ImageNet trained CNN. Having a CNN trained on a brain-related datasets (e.g., BraTS datasets) could potentially provide better performance metrics. Ideally, a comparison between two transfer-learned approaches with different initial training (imageNet vs. BraTS) needs to be provided.
  • A rare brain examination is performed without T2-weighted imaging, particularly, a TSE-T2-weighted imaging, which is intrinsically a black-blood sequence. Thus, any examination would already have a black-blood image that could serve as an additional CNN data input. Therefore, a direct T1-CE vs BB comparison is far from practical application, a T1-CE + T2-TSE vs BB + T2-TSE would provide a more practical Alternatively, pre-contrast T1-weighted images, particularly with flow-related enhancement, might serve as a vessel indicator and assist a CNN in decision-making.
  • There is a significant difference in scanning parameters of T1-weighted images and BB images (matrix, resolution, TR), that could influence, for example image SNR. AN increase in performance is then not directly indicative of BB being a better contrast for metastasis detection. A comparison between sequences with similar acquisition protocols is required.
  • There is not enough information on the acquisition pulse sequences in the methods section. What was the T1-CE sequence? Is the BB-sequence IR- or TSE-based? What are the slice thicknesses in both sequences?

Author Response

Dear reviewer,

please find our response in the attached word document.

We thank you for the excellent and professional review of our work!

Reviewer 2 Report

Congratulations. Really nice work.

A couple of writing questions:

  • Line 102 there´s a lack of some "0" values. They appear as .39 or .49, while in the rest of the text you have 0.5, 3.7 and so.
  • Line 156 you repeat "in the" at the beginning of the sentence.
  • Line 237, reference 6 is not in the same format as the rest of them.

And some technical and procedural questions:

  • I'm not an engineer but an imaging professional. Just to let you know my background. All my doubts and comments are more focused on the applicability of this protocol to medical imaging.
  • How did you decide the inclusion criteria in "negative" or "positive" groups? Is based on a medical diagnosis? Who made this diagnosis? It would be nice to have this information in "patient population" section.
  • In the same way, do you have the sizes of the lesions? You could indicate the range of these values and their frequency. This could help the reader about the tumour sizes you deal with.
  • I have some doubts regarding the conclusions with the small metastasis and the different pixel size of the images. Why do you have this difference? Could it affect this difference to the sensitivity of both methods when they find a small lesion? Attending to your information, there is a difference around 25% between pixel sizes, quite considerable when dealing with small structures. Again, the lesion size is a key data that should be incorporated to the information.
  • At the end, how do you decide which one is the best protocol? Depending on the sensitivity and sensibility, but how? I feel a bit messy in the text at this point. The system compares the positive studies to the negative controls, ok. And then, what happen? How do you evaluate the accuracy of the protocol? Comparing the number of detected lesions or just the positive correlation between health/illness? The size of these lesions? That´s really important if you want to implement this tool in medical automatic diagnosis.

That´s all. Really a nice work.

Author Response

(The authors gave the same response as above.)

Round 2

Reviewer 1 Report

While very minor changes have been introduced into the manuscript, the authors need to address a number of critical points, some noted by both reviewers in the first review round and addressed properly in the preceding works, particularly:

- A difference in slice thickness suggests the black-blood image to be a 3D-encoded pulse sequence, whereas CE-T1 seems to be a 2D pulse sequence, although reported matrix sizes suggest otherwise. Please clarify this.

- A difference in encoding process (i.e., 2D versus 3D) could be a critical issue for the performance comparison of the two datasets, as there would be some data missing from the CE-T1 dataset due to the inter-slice gap and increased slice thickness, there would be a different set of artifacts in the two (e.g., due to slice cross-talk), there would be different SNR in the two datasets, leading to complicated interplay between detection using the two types of images. The favorability of one dataset over another would not be as simple as the authors imply in their response. A unification procedure would also be unable to recover missing data from the CE-T1 dataset. I would strongly suggest the research to be re-done with proper imaging parameters. Another option is a meticulous resolution unification, taking into account at minimum slice thickness, slice gaps and resolution difference. This must be described in detail in the methods section.

- Limitations of the study rising from a comparison of different image resolutions should be analyzed and explicitly stated in the manuscript (please, note that the work, that initially demonstrated a CNN for black-blood image detection, indeed uses dame voxel sizes).

- The authors refuse to include the T2-based dataset into their work, resulting in their work differing very little from the cited work of Yang. A combination with an intrinsically black-blood protocol can provide synthetic black-blood images (as in https://www.nature.com/articles/s41598-018-27742-1). A comparison of CNN performance on synthetic and acquired black-blood images would be of much higher interest.

- A performance comparison with different CNNs cited in the text would resolve the question of if the observed increase in the metastases detection efficiency is a consequence of a better pulse sequence choice (as claimed by the authors) or merely of an poorly match of the CNN architecture to the data.

- Exact pulse sequence names should be provided

- For TSE-based sequences echo train length should be reported

- Define all abbreviations prior to their first use (e.g., SAD, CE), also the same abbreviation (CE) is used in two different meanings in the manuscript.

Author Response

We thank the reviewer for his or her thorough and expert review of our work. Please find our answers attached.
